# Estimating the Rates of Acquisition and loss of Resistance of *Enterobacteriaceae* to Antimicrobial Drugs in Pre-Weaned Dairy Calves

**DOI:** 10.3390/microorganisms9102103

**Published:** 2021-10-06

**Authors:** Elisa Cella, Emmanuel Okello, Paul V. Rossitto, Beniamino T. Cenci-Goga, Luca Grispoldi, Deniece R. Williams, David B. Sheedy, Richard Pereira, Betsy M. Karle, Terry W. Lehenbauer, Sharif S. Aly

**Affiliations:** 1Veterinary Medicine Teaching and Research Center, School of Veterinary Medicine, University of California Davis, Tulare, CA 93274, USA; ecella@ucdavis.edu (E.C.); eokello@ucdavis.edu (E.O.); paulrossitto799@gmail.com (P.V.R.); dvmwilliams@ucdavis.edu (D.R.W.); dbsheedy@ucdavis.edu (D.B.S.); tlehenbauer@vmtrc.ucdavis.edu (T.W.L.); 2Department of Population Health & Reproduction, School of Veterinary Medicine, University of California Davis, Davis, CA 95616, USA; rvpereira@ucdavis.edu; 3Laboratorio di Ispezione degli Alimenti di Origine Animale, Dipartimento di Medicina Veterinaria, Università degli Studi di Perugia, 06126 Perugia, Italy; beniamino.cencigoga@unipg.it (B.T.C.-G.); grisluca@outlook.it (L.G.); 4Cooperative Extension, Division of Agriculture and Natural Resources, University of California, Orland, CA 95963, USA; bmkarle@ucanr.edu

**Keywords:** antimicrobial resistance, dairy calves, ceftiofur, neomycin, feces

## Abstract

The objective of this study was to investigate the effect of the antimicrobial drugs (AMD) on the shedding of resistant Enterobacteriaceae in feces of pre-weaned dairy calves. The AMD considered were ceftiofur, administered parenterally, and neomycin sulfate added in milk replacer and fed to calves during the first 20 days of life. Fifty-five calves, aged one to three days, were enrolled and followed to 64 days. Fecal samples were collected three times/week and treatments recorded daily. Enterobacteriaceae were quantified for a subset of 33 calves using spiral plating on plain, ceftiofur supplemented, and neomycin supplemented MacConkey agar. Negative binomial models were used to predict the association between treatment with AMD and the gain and loss of Enterobacteriaceae resistance over time. Acquisition of resistance by the Enterobacteriaceae occurred during treatment and peaked between days three to four post-treatment before decreasing to below treatment levels at days seven to eight post-treatment. Acquisition of neomycin resistance was observed on the first sampling day (day four from the start of feeding medicated milk replacer) to day eight, followed by cyclical peaks until day 29, when the Enterobacteriaceae counts decreased below pre-treatment. Enterobacteriaceae resistance against both AMD increased after AMD administration and didn’t return to pre-therapeutic status until seven or more days after therapy had been discontinued. The study findings provide valuable insights into the dynamics of Enterobacteriaceae under routine AMD use in calves.

## 1. Introduction

Antimicrobial resistance (AMR) of bacteria is a substantial threat to both animal and human health [1]. The evolution and spread of AMR are associated with the use and overuse of antimicrobial drugs (AMD) [2]. Over the last 50 years, the number of species and strains of pathogenic and commensal bacteria resistant to AMD and the number of AMD to which they are resistant, including resistance to three or more drug classes, also known as multi-drug resistance (MDR), has greatly increased worldwide [2]. An example is *Escherichia coli*, which are ubiquitous in the feces of animals and humans [3,4]. Although commensal *E. coli* strains seldom cause infections, they can harbor AMR genes that may be transferred to other pathogenic or opportunistic bacteria, leading to intractable infections [3,5,6].

However, several studies have demonstrated that the presence of AMR in *E. coli* from calf feces are associated with calf age and are more abundant in pre-weaned calves [7,8] where *E. coli* is resistant to multiple antimicrobials [9,10]. The higher incidence of AMR in pre-weaned dairy calves may be due to their high susceptibility to diseases, and hence their increased exposure to AMD for the treatment of these diseases. Moreover, the high prevalence of AMR in *E. coli* in dairy calves can be a source of AMR genes for other bacteria that share the same environment [11]. The frequent and prolonged use of AMD in individual calves has been associated with the increased shedding of resistant *E. coli* [11,12].

Feeding hospital (waste) milk to calves is a common practice on many dairy farms [13,14]. Waste milk is non-saleable milk either from cows transitioning from colostrum to milk or from cows that were treated with medication with a milk residue withholding period [15,16]. Randall et al. [16] reported that, in the UK, waste milk contained third or fourth generation cephalosporin residues which could potentially exert a selective pressure influencing the occurrence of extended spectrum β-lactamase resistant bacteria in calves fed waste milk [16]. Berge et al. reported the prevalence of Cephalothin and Cefotaxime resistant *E. coli* in bulk milk of 6% and in waste milk of 7% [17]. In their study, Berge et al. [17] followed up calves from birth to four weeks of age, however, fecal samples were collected every two weeks (on day 1, 14, and 28). A greater frequency of sampling is required to model the impact of ceftiofur on AMR in fecal *E. coli* from calves.

The practice of feeding waste milk to young calves may impose selection pressure on the calves’ gastrointestinal bacteria and favor resistant strains [18] which may explain the high prevalence of AMR *E. coli* in pre-weaned calves. Prophylactic use of AMD, such as neomycin, in milk replacer or whole milk fed to calves has been reported. Urie et al. [19] found that 38% of heifer calves in the United States suffer from disease, with 56% of those calves experiencing diarrhea and 33% experiencing respiratory illness. Specifically, 56% of all United States dairy operations reported feeding medicated milk replacer to calves [20]. A mixture of neomycin and oxytetracycline is commonly used, with 9% of dairy operations relying on this combination as the main antimicrobial additive [21]. Okello et al. surveyed California dairies on AMD usage and reported that neomycin was one of the most common calf AMD used during the pre-weaning period; specifically, 62.8% of the respondents received neomycin sulfate in milk or milk replacer [21]. In addition, 16.3% of the survey respondents indicated that ceftiofur was the first AMD choice to treat respiratory diseases and 6.7% used it to treat diarrhea in pre-weaned calves [21].

Hence the aim of the current study was to model the dynamics of AMR post-AMD use in pre-weaning dairy calves followed from birth to weaning. The primary objective of this study was to investigate the rates at which AMR in Enterobacteriaceae are acquired and lost under routine management of pre-weaned dairy calves exposed to ceftiofur and neomycin during the pre-weaning period.

## 2. Materials and Methods

### 2.1. Study Animals

The study protocol was approved by the University of California Davis Institutional Animal Care and Use Committee (Protocol 19871). Fifty-five calves on a calf ranch located in the Southern San Joaquin Valley of California were enrolled in December 2017 and followed from arrival (one to two days of age) at the premises to weaning (approximately 64 days of age). All the calves that arrived at the calf ranch on the first day of the study were enrolled on the same day. The calves were sourced from six different local dairy farms. Due to the difference in time of birth for individual calves and the transportation schedules for daily pickup from the various dairies, calves delivered to the calf ranch ranged from 1 to 2 days old at enrollment.

Colostrum from all six source dairies was delivered along with the calves to the calf ranch where the colostrum was pooled and fed to calves for the first two days from arrival (3 L/day). Three of the source dairies treated cows at dry-off with an intramammary infusion of Spectramast^®^ DC (Zoetis, Kalamazoo, MI, USA), one dairy treated dry cows with Orbenin-DC (Intervet/Merck Animal Health, Madison, NJ, USA), the fifth dairy treated cows at dry-off with an AMD intramammary infusion that was not identified, and the sixth dairy did not treat cows at dry-off. Calves were individually housed in wooden hutches with slatted sides which were placed on concrete slabs and had a front metal panel design with spaces that allowed nose-to-nose contact between adjacent calves. All the calves were fed twice daily, 1.89 L of milk replacer supplemented with vitamins and minerals (Forticalf, Agri-Best, Strauss Feeds, Watertown, WI, USA) and neomycin sulfate (Neosol, Med-Pharmex Inc., Pomona, CA, USA) according to a veterinary feed directive (FDA) [22] for the first 20 days as an extra-label drug use provision due to the maximum duration of 14 days listed on the product label (10 mg/pound of body weight/day). Subsequently, calves were fed pasteurized waste milk (3l twice a day) from 21 days old to the end of the sample collection period at 64 days of age. Waste milk was delivered from the calves’ source dairies. Starter feed and water were made available to the calves ad libitum.

### 2.2. Treatment Data

Given the nature of the current study being observational, specifically a cohort study, treatments were administered to the calves based on the treatment protocol established by the herd veterinarian, and the research team had no role in disease diagnosis or treatment decisions. Initial assessment of hutched calves prior to start of the study showed that ceftiofur hydrochloride was the most common AMD administered, and neomycin was added to the milk replacer.

The AMD therapies were administered by designated calf ranch staff. Treatment information for each enrolled calf was logged onto a hutch card attached to the front of each hutch. Additional information recorded on the hutch cards included the calf identification (ID), study enrollment ID, date and type of treatment and the initials of the person administering the treatment. Treatment data including AMD treatment dates, calves’ birth dates and source dairies, sampling information including sample collection dates, and fecal score were collected and maintained in a study database.

### 2.3. Fecal Sample Collection

Fecal samples were collected from all 55 calves at enrollment and thereafter three times every week (Monday/Wednesday/Friday). Fecal samples were collected per rectum by digital manipulation using a clean disposable latex examination glove overlayed with sterile lubricant (Priority Care, Elgin, IL, USA) and transferred into 50 mL polypropylene tubes. Study personnel used coveralls for personal protection and biocontainment when entering each calf hutch, the coveralls were changed after exiting the calf stall if they were contaminated with fecal matter. To prevent cross-contamination, disposable plastic boots, palpation sleeves and gloves were changed between calves. In addition, the study staff scored each calf’s fecal consistency output at sampling as 1 (normal), 2 (loose) or 3 (watery).

### 2.4. Quantifying Antimicrobial Resistant Enterobacteriaceae

Fecal samples were transported immediately on wet ice to the Dairy Epi Lab (Aly Lab) at the Veterinary Medicine Teaching and Research Center (VMTRC) for same-day processing and storage. For each fecal sample, a total of 0.8 g of feces were weighed and transferred into a sterile snap-cap polypropylene microcentrifuge tube and homogenized in an 800 µL of tryptone soy broth (TSB) in 50% glycerol (25% glycerol final concentration). The processed samples were immediately stored at −80 °C.

Due to funding limitations, a random subset of 33 calves out of the study’s 55 enrolled was identified using a random number generator (Excel, Microsoft, Redmond, WA, USA) for fecal Enterobacteriaceae quantification. The 33 calves were stratified by treatment status (13 ceftiofur and neomycin treated calves and 20 neomycin treated calves) to quantify antimicrobial resistant Enterobacteriaceae.

Antimicrobial resistance was determined for the two most administered AMD on the study calf ranch. Antimicrobial resistance of Enterobacteriaceae was tested using spiral plating (Spiral System Inc., Cincinnati, OH, USA) on MacConkey agar plates supplemented with either ceftiofur hydrochloride (Alfa Aesar, Ward Hill, MA 01835, USA) or neomycin sulfate (Alfa Aesar, Ward Hill, MA 01835, USA). Internal quality control strains used were *E. coli* ATCC 25,922 (American Type Culture Collection, Manassas, VA 20108, USA) as a susceptible strain and *E. coli* UCD-DEL-7B (tested at the UC Davis School of Veterinary Medicine Clinical Pathology Laboratory with resistance >8 µL/mL for ceftiofur and >32 µL/mL for neomycin; wild type isolate from Dairy Epidemiology Laboratory, DEL; Veterinary Medicine Teaching and Research Center, School of Veterinary Medicine, University of California Davis, Tulare, CA 93274, USA) as a resistant reference strain.

For the validation of the concentrations at which AMD were supplemented in the MacConkey agar, the two reference *E. coli* strains, ATCC 25,922 and UCD-DEL-7B, were plated at a concentration of 10^7^ CFU/g on MacConkey agar supplemented with 10-fold dilution concentrations of ceftiofur (30, 3, 0.3, 0.003 µg/mL) and neomycin (800, 80, 8, 0.8, 0.008 µg/mL), respectively. Specifically, *E. coli* concentration was determined by measuring the optical density using a spectrophotometer. The two reference *E. coli* strains were then plated onto sheep blood agar (Hardy Diagnostics, Santa Maria, CA 93455) and incubated overnight at 37 °C. Single colonies were resuspended in TSB (OXOID, Basingstoke, Hampshire, England), the bacterial concentration estimated by measuring the optical density (600 nm) and adjusted to 1 × 10^7^ CFU/g in TSB. Ten-fold serial dilutions were prepared using a phosphate buffered saline (PBS) (Fisher BioReagents, Fair Lawn, New Jersey 07410) and the dilutions 10^−3^ and 10^−4^ were spiral plated on MacConkey agar with and without the different concentrations of AMD added. Clinical and Laboratory Standards Institute (CLSI) breakpoints based on clinical outcomes from animals were not used; instead the final AMD concentration for each of the study AMD was selected based on the higher concentration at which the reference strains were inhibited in the validation experiment. The final MacConkey infused AMD drug concentrations were 30 µg/mL and 50 µg/mL for ceftiofur and neomycin, respectively.

Quantification of total resistant and susceptible Enterobacteriaceae populations was performed by spiral plating serial dilutions of the fecal samples on plain and AMD infused agar plates. Fecal samples were thawed from −80 °C at room temperature and resuspended in 3.4 mL of PBS and serial dilutions from 10^−1^ to 10^−4^ were prepared. For each sample, 50 µL from the 10^−2^ and 10^−4^ dilutions were plated in duplicate using the spiral plating machine. Each sample was plated on plain MacConkey agar, MacConkey agar supplemented with ceftiofur 30 µg/mL and MacConkey agar supplemented with neomycin 50 µg/mL. The plates were incubated overnight (14–18 h) at 37 °C and the colonies were counted using the counting grid. Total plate count was conducted if less than 20 colonies were observed in the four segments of the wedge counted. Samples that had higher Enterobacteriaceae concentration in MacConkey supplemented with antibiotics compared to the same sample’s plain MacConkey plates were re-cultured.

Culture results were expressed as count per plate (and CFU/g of feces). Two dilutions were plated (10^−2^ and 10^−4^) for each sample and the average CFU/g of the duplicate plates was calculated. Only one dilution was selected and its average was used for the data analyses; the dilution selection was based on the countable plates. The presence of zero growth for a dilution or the presence of >300 colonies/plate (defined as the maximum countable colonies/plate), were considered as exclusion criteria for that dilution and the average CFU/g of the countable dilution was selected. In presence of no growth in both dilutions, zero colony growth was used for the analyses. In presence of countable colonies in both dilutions’ duplicate plates, the dilution that presented the highest number of colonies per plate was selected and the respective mean CFU/g was used for the analyses.

### 2.5. Statistical Analyses

#### 2.5.1. Enumerating Ceftiofur Resistant (≥30 µg/mL) and Sensitive (<30 µg/mL) Enterobacteriaceae

The analyses investigated the associations between treatment, source dairy, and fecal score at sampling, duration of treatment and the outcome concentration of Enterobacteriaceae resistant to ceftiofur at ≥30 µg/mL. The concentration of Enterobacteriaceae as measured in average CFU/g feces was assessed using a mixed effects negative binomial model with a random intercept for calf and a random covariate (slope) for time to account for the repeated measures within calf. The six source dairies of the enrolled calves were recorded, and a categorical variable was created based on the source dairy of the calves.

The Duration Post-AMD Administration Time (DPAT) variable was designed to specify both ceftiofur treatment status and repeated measures over time for the study calves. Specifically, the entire duration of follow up for untreated calves and duration of follow up prior to treatment for treated calves were coded as 0 in the DPAT variable. The duration of time during which an AMD treatment regimen was administered to treated calves was coded as 1. However, post-treatment DPAT was coded as the actual days elapsed from after the final treatment administered and end of follow up (age 64 days).

Due to calves sampled three times a week only, transformations of DPAT were explored. In addition to the actual days elapsed post-treatment as specified above, a form of DPAT with days elapsed post-treatment grouped into 2, 3, 4, 5, 6 and 7-day periods from last treatment administered to end of follow up was specified. The categorical DPAT coefficients, therefore, estimated the difference between the fecal Enterobacteriaceae concentration in samples from untreated calves and prior to treatment in ceftiofur treated calves (DPAT 0, the referent), and concentration in samples during the treatment administration (DPAT 1), and the period post-treatment (DPAT 2, 3, 4, 5, etc.).

Furthermore, the associations between treatment, source dairy, fecal score at sampling, duration of treatment and the outcome Enterobacteriaceae sensitive to ceftiofur at <30 µg/mL (average CFU of Enterobacteriaceae/g feces/calf/sample) was modeled using a similar mixed effects negative binomial model as described above. Specifically, the concentration of Enterobacteriaceae sensitive to ceftiofur at <30 µg/mL (average CFU of Enterobacteriaceae/g feces) was calculated as the difference between the average CFU growth on plain MacConkey plates and the average CFU growth on the ceftiofur impregnated MacConkey plates.

In addition, the association between ceftiofur treatment and Enterobacteriaceae sensitive to ceftiofur at the <30 µg/mL concentration was modeled using a mixed effects negative binomial model. The counts were estimated as the arithmetic difference between the average CFU/g of Enterobacteriaceae counts on plain MacConkey and average CFU/g of Enterobacteriaceae on MacConkey infused with ceftiofur. The random effects included in the model were calf as a random intercept and the calf’s age as a random covariate (slope).

#### 2.5.2. Enumerating Neomycin Resistant (≥50 µg/mL) and Sensitive (<50 µg/mL) Enterobacteriaceae

For the neomycin analysis, a negative binomial mixed model was specified with a similar random effect structure as in the ceftiofur resistance model described above with the outcome Enterobacteriaceae resistant to neomycin at ≥50 µg/mL (average CFU of Enterobacteriaceae/g feces/calf/sample). Since all the study calves were treated for the first 20 days of life, the treatment exposure over time was specified first as a DPAT; however, this variable had no pre-treatment dummy variable. In addition, an alternate specification for the exposure to treatment was modeled as Sampling Day (SD). For both DPAT and SD, the variable levels were specified in one day categories, or grouped in 2, 3, 4, 5, 6 or 7 day-period intervals with the respective coefficients representing the first 20 days from arrival to the calf ranch interpreted as during neomycin treatment, and the subsequent coefficients as post-treatment.

A negative binomial model for neomycin sensitive Enterobacteriaceae (<50 µg/mL) could not be specified due to the difference in the Enterobacteriaceae counts in the neomycin infused and non-infused MacConkey including negative counts.

For all models, a manual model building approach was utilized using likelihood ratio tests to compare nested models and Akaike’s information criterion (AIC) to assess model goodness of fit. Furthermore, interaction terms between each of the variable’s duration of treatment, fecal score, source dairy and the variable DPAT were explored and confounding by dairy and duration of treatment assessed using the method of change in estimates as described elsewhere [23]. A *p*-value ≤ 0.05 was considered statistically significant. All statistical analyses were conducted in Stata IC 16 (Stata Statistical Software, College Station, TX, USA).

## 3. Results

The overall incidence of ceftiofur treatments in the 55 enrolled calves was 58% (32 of 55). Estimates reported here onwards refer to the 33 calves randomly selected for bacteriological testing. Twenty four calves were two days old and nine were one day old at enrollment. A total of 849 fecal samples were collected over the study period; nine samples were not collected due to empty rectum at the time of collection on some days. During the study period, among the 33 animals enrolled, 11 were treated with ceftiofur (Table 1). The remaining 20 study calves did not receive any injectable AMD over the study period.

### 3.1. Ceftiofur Antimicrobial Resistance Model (>30 µg/mL)

The model with the categorical DPAT variable with variable levels in one-day increments after the last treatment in ceftiofur treated calves did not converge. Models with DPAT grouped in 3, 4, 5, 6 and 7 day-periods post-treatment to end of follow up (DPAT 3, 4, 5, 6, 7) showed no significant differences in Enterobacteriaceae concentration between the follow up times post-treatment compared to DPAT 0 (reference). In contrast, the model with DPAT specified in 2 day-period intervals post-treatment had better model fit (lowest AIC compared to the other models) with the coefficient comparing the counts from days 2 and 3 post-treatment significantly higher than DPAT 0. Further grouping of the non-significant coefficients showed that the best fitting model specified 2 days period intervals for up to 14 days post-treatment with the remaining follow up period days 22 to 64 as a single coefficient (Table 2). Fecal score was retained in the final model due to its importance, and despite being non-significant (*p* = 0.062), adding it resulted in better model fit (lower AIC). One of the two-way interactions was significant and no evidence of confounding was observed using a 20% change in estimates criterion.

Figure 1 shows the CFU/g of Enterobacteriaceae resistant to ceftiofur 30 µg/mL in untreated calves and calves treated with ceftiofur during the first nine observation days. The x axis shows the age in days centered around the last treatment. The figure shows an increase in CFU/g of Enterobacteriaceae in the ceftiofur treated animals post-treatment, which persists into the following days and eventually declines, while in untreated calves the concentration of Enterobacteriaceae decreases starting on the second day of the study but remains <5 × 10^6^ CFU/g. Figure 2 depicts a similar trend of Enterobacteriaceae concentration in calves that received the ceftiofur treatments 10 to 18 days from enrollment. The peak of resistance corresponds to the calves’ treatment period followed by a decreasing concentration of resistant Enterobacteriaceae in the post-treatment period. Specifically, Enterobacteriaceae concentration started to decrease on day—4 with a dramatical reduction from day 0 (last day of treatment) to day 4 post treatment until reaching a significantly low concentration at approximately day 22 post treatment compared to the last day of treatment. Figure 3 shows the Enterobacteriaceae concentration trend for untreated calves and calves treated with ceftiofur 19 days or more from enrollment. The treated calves Enterobacteriaceae concentration increased during the treatments but remained below 5 × 10^6^ CFU/g followed by a decrease under 1 million CFU/g after six days from the last treatment until the end of the study. Untreated calves had a similar trend in Enterobacteriaceae counts to treated calves in Figure 1, Figure 2 and Figure 3 within a week from last treatment.

### 3.2. Ceftiofur Sensitive Enterobacteriaceae (<30 µg/mL)

Both dairy source and fecal score variables were not significant resulting in the model with DPAT having the lowest AIC and better model fit (Table 3). The variable DPAT’s reference was Enterobacteriaceae concentration sensitive to ceftiofur < 30µg/mL as observed in untreated calves during the entire study period and in the treated calves before treatment with ceftiofur. The model showed that there was a numerical increase in the concentration of ceftiofur sensitive (<30 µg/mL) Enterobacteriaceae during ceftiofur treatment. Subsequently, at days 5–6 post treatment there was a numerical decrease in the concentration of Enterobacteriaceae sensitive to ceftiofur < 30 µg/mL. The decrease in CFU/g continued on days 7–8, 11–12 and 21–58 after treatment (Figure 4).

### 3.3. Neomycin Antimicrobial Resistance Model (>50 µg/mL)

Dairy source and fecal score variables were excluded due to the non-significant variables. The final model is summarized in Table 4, and shows an overall increase in CFU/g on days four to eight, 14 and 18 interlaced with a decrease on days 11, 15 and day 20, producing a cyclical pattern. Subsequently, there were no significant differences in CFU/g between day 22 and 37 compared to the first day of follow up. For the remaining period there was a significant decrease in CFU/g compared to the first day of follow up. Figure 5 depicts the neomycin resistant (50 µg/mL) Enterobacteriaceae concentration increase after the first shot followed by a peak on day 6. Subsequently, there is a decrease in Enterobacteriaceae concentration below 20 × 10^6^ CFU/mL on the last day of treatment (day 20) and a constant decrease until day 64.

Figure 6 depicts the difference in the concentration of Enterobacteriaceae between MacConkey non-infused and MacConkey infused with 50 µg/mL of neomycin. An initial concentration of neomycin sensitive (50 µg/mL) Enterobacteriaceae at 75 CFU/g of feces (higher concentration of Enterobacteriaceae on the MacConkey agar non-infused with antibiotic compared to the concentration on the neomycin infused agar) continued to decrease to 0 CFU/g at day six of follow up (higher concentration of Enterobacteriaceae on the neomycin infused agar than the concentration on the MacConkey non-infused). After day 6 the concentration of Enterobacteriaceae of neomycin MacConkey agar continued to be greater than the Enterobacteriaceae on the MacConkey non-infused until day 34 of follow up, hence the negative sign for the difference in counts of Enterobacteriaceae in neomycin infused versus non-infused MacConkey. From day 34 to the end of the study there was a decrease in neomycin resistant Enterobacteriaceae, resulting in the presence of neomycin sensitive (50 µg/mL) Enterobacteriaceae that remained below 1 × 10^6^ CFU/g of feces.

To facilitate interpretation of the mixed negative binomial models, the non-linear combination estimates for each model’s predicted colony counts are summarized in Table 5, Table 6 and Table 7.

## 4. Discussion

The current study investigated the effect of antimicrobial treatment on the shedding of resistant Enterobacteriaceae in feces of pre-weaned dairy calves administered ceftiofur as a systemic injection for disease treatment, and neomycin sulfate administered in milk replacer for the first 20 days of the study as an oral solution for treatment and control of diarrhea. Given that neomycin sulfate oral solution is administered in water, milk, or milk replacer and is not classified as a veterinary feed directive drug, this medically important antimicrobial can be used in an extra-label manner with a veterinary prescription or authorization by a licensed veterinarian. In this project, neomycin administration for the first 20 days was greater than the label information that indicated a maximum administration of 14 days. An authorizing veterinarian should follow good antimicrobial stewardship principles (AVMA) [24] in making these decisions and should be clear about the medical evidence or justification for such extra-label uses of medically important antimicrobials.

Antimicrobials such as neomycin may be added to feed such as milk or milk replacer to prevent disease [24,25]. However, antimicrobials create a selection pressure for antimicrobial-resistant bacteria [16]. Berge et al. [17] demonstrated that fecal pathogens shed by pre-weaned calves show an increase in AMR when the animals are exposed to prophylactic treatments with AMD, particularly ones added to milk replacer. A study conducted in Minnesota [26] showed similar results of transient resistance as this study does. Foutz et al. [26] evaluated the AMR score (given to the isolate according to the number of antimicrobial classes to which it was resistant) between calves and observed that calves fed with medicated milk replacer had a higher AMR score than the calves fed unmedicated milk replacer. The resistance was transient resolving after weaning (at week 16), suggesting that once the antimicrobial pressure was removed there was a re-establishment of the susceptible *E. coli* bacteria [26]. An important aspect that we could not evaluate in our study was the comparison between calves treated and untreated with neomycin, since all the animals were treated with this antimicrobial.

In the current study, pre-weaned calves received prophylactic and therapeutic AMD. Respiratory disease and diarrhea are the most common diseases that affect young calves and the most common cause for AMD use in this age group [27]. Diarrhea was the treatment indication for the ceftiofur treatments administered to the study’s 13 calves. The U.S. Food and Drug Administration (FDA) defines the posology and limitations of AMD usage in the U.S. Ceftiofur is an extended-spectrum cephalosporin that has been approved for therapeutic use in cattle in the U.S. since 1988. Ceftiofur is indicated for the treatment of respiratory tract infections, metritis, and foot rot. The administration of ceftiofur to treat diarrhea in calves in the U.S. constitutes extra label drug use and requires the approval of the herd veterinarian [11].

The Antimicrobial Animal Drug Distribution Report 2019 (https://www.fda.gov/media/144427/download; Accessed 30 September 2021) reported that cephalosporins ranked third in terms of antimicrobials sold in the U.S. during that year at 29,830 kg. In contrast, the European Union (EU) regulated the use of ceftiofur in food-producing animals in 2009 (regulation n. 37/2010). The EU regulation included the maximum residue limits (MRL) of pharmacologically active substances which for ceftiofur included the MRL in muscle, fat, liver, kidney and milk. Although data collection on use of veterinary antimicrobials in the EU is not compulsory, it is regulated by national legislation and/or obtained on a voluntary basis. The European Surveillance of Veterinary Antimicrobials Consumption (ESVAC) is a system in place that collects and tracks the sales of veterinary antimicrobials [28]. The ESVAC system reports sales of antimicrobial drugs per population correction units (PCU). The PCU is an estimated biomass calculated from the weight at treatment of livestock and of slaughtered animals in a given year and is used to correct the antimicrobial consumption (in mg) for the animal population at risk of being treated with antimicrobials (in kg). For example, ESVAC reported a range between 0.4 mg/PCU of sales of the third and fourth generation of cephalosporins in the period from 2010 to 2018 in Italy. Comparing such an estimate to US reported sales would require multiplying by the total estimated PCU (in tonnes). Hence, assuming the estimate of 0.4 mg/PCU and the ESVAC-reported PCU for the same country as 3819.3 tons results in an estimated sales for cephalosporins in Italy in 2018 of 1.53 tonnes (0.4 mg/PCU * 3819.3 mg/PCU/1000 tons = 1527.72/1000 = 1.53 tons) of third and fourth generation cephalosporins. Furthermore, within Italy, the sales of third and fourth generation cephalosporins in 2018 were lower compared to other antimicrobials such as tetracyclines (150 mg/PCU). The FDA proposed a similar metric based on biomass which would facilitate comparison of animal antimicrobial sales between states and countries [29].

Previous studies reported different levels of *E. coli* ceftiofur resistance in calves. White et al. [30] reported a prevalence rate of 69% of ceftiofur-resistant *E. coli* isolates from preweaning calves from which fecal samples were collected once every two weeks for the first 60 days of life. Another study that examined the antimicrobial susceptibility of *E. coli* from feces of calves with diarrhea, or intestinal tissue from septicemic calves taken postmortem, where all the isolates represented cases that had failed antimicrobial therapy reported a prevalence rate of 13% of ceftiofur resistant *E. coli* [31].

In the current study we observed ceftiofur resistance (≥30 µg/mL) in Enterobacteriaceae at enrollment which continued to decline in all the study calves except those treated with ceftiofur (Figure 1). The initial ceftiofur resistance in Enterobacteriaceae after birth (one to two days of age) may be due to the colostrum fed to calves containing residues of dry-cow antimicrobial drugs, followed by a decrease in CFU/g in the following two days. Alternatively, the initial resistance peak after birth, in untreated calves and prior to any treatments in the treated calves, could be due to environmental bacteria ingested by the calves and colonizing their gut microbiota. Most of the treatments were administered between day six and day nine, which corresponded to the peak of resistance observed in ceftiofur treated calves (Figure 1).

The significant increase in CFU/g associated with ceftiofur treatments on days three to four after treatment (Table 2), suggests that the acquisition of resistance can be quantified within four days after the treatment, before a decrease in Enterobacteriaceae counts on ceftiofur infused MacConkey plates is detected. Such a resistance pattern indicates that the acquisition and loss of resistance is associated with the antimicrobial treatments administered to the animals. A similar study conducted in adult dairy cows reported a peak of resistance during treatment and a return to pre-treatment levels seven to eight days post-treatment [32]. Singer et al. [33] assessed the effects of systemic treatment with ceftiofur in adult cows on antimicrobial resistance patterns of fecal *E. coli* and reported that before treatment, fecal *E. coli* were susceptible to ceftiofur. However, immediately following treatment and during the treatment protocol there was a decrease in bacterial counts (decrease in the susceptible population) and an increase in resistance. Singer et al. [33] also reported a decrease in resistance by the seventh day after the last treatment after which the sensitive *E. coli* started to flourish again; such a transient effect could hold true with respect to calves exposed to antimicrobials as well [33].

Our neomycin sensitive (50 µg/mL) Enterobacteriaceae graph depicts an initial high concentration followed by a decrease below 0 CFU/g due to the presence of resistant Enterobacteriaceae. Specifically, after five days from initiation of treatment with neomycin, a higher growth of Enterobacteriaceae was observed on MacConkey infused with neomycin (50 µg/mL) compared to MacConkey not infused with neomycin. The reason for the presence of higher growth of neomycin resistant Enterobacteriaceae compared to the growth on non-AMD infused media is currently unknown. Contrary to the estimate of Enterobacteriaceae sensitive to ceftiofur (<30 µg/mL), we were not able to model Enterobacteriaceae sensitive to neomycin (<50 µg/mL) given that the range of the difference between counts included negative values, a violation of the negative binomial mixed model. Future studies will be focused on resistance genotypes to study discrepancies between phenotypic and genotypic evidence of antimicrobial resistance.

Despite the reduction in the number of calves studied (33 of the 55 enrolled), there was no loss of power, as evident from the statistically significant differences in each of the study models. One of the limitations of the current study was the Enterobacteriaceae family-level identification of commensal isolates. Further speciation of the study colonies would not have been informative beyond the finite number of colonies that could have been speciated, given that AMR was determined for the entire culturable population of Enterobacteriaceae. Hence, without a prohibitive number of isolates speciated across all the study calves and timepoints, speciation would not have informed the percent of study Enterobacteriaceae that were *E. coli*. Furthermore, use of a more selective media than MacConkey would have been cost-prohibitive given that the study included more than 10,000 plates.

Another study limitation was that concentrations of AMDs infused into the MacConkey agar were not based on CLSI guidelines, but rather on the in vitro validation experiment of reference strains. The CLSI recommended breakpoint was not assessed in the validation experiment amongst the ceftiofur concentrations explored (0.03, 0.3, 3 and 30 µg/mL). The lack of growth inhibition on MacConkey infused with all but the highest ceftiofur concentrations rendered a contrast with Enterobacteriaceae growth on plain MacConkey not possible. Hence, the final breakpoint used for ceftiofur was higher than the CLSI recommendation to determine AMR in *E. coli* isolated from clinical cases (30 µg/mL versus 8 µg/mL). The higher breakpoint shifted the interpretation of the impact of ceftiofur treatment on AMR in *E. coli* to resistance at higher concentrations than the clinical breakpoint. The reason for the lack of inhibition of Enterobacteriaceae from our calf samples at concentrations lower than 30 µg/mL on MacConkey agar are not known; however, this contrasts with our earlier report on the impact of ceftiofur treatment on Enterobacteriaceae from adult cows where growth inhibition was observed at concentrations as low as 1 µg/mL ceftiofur-infused MacConkey agar [32].

Future research should explore calves’ resistome changes during early life and the taxonomic profiles investigated using 16S rRNA sequencing. To characterize the dynamics of resistome over time, shotgun metagenomic sequencing should be employed. Specifically, the genes that code for resistance against neomycin and ceftiofur in Enterobacteriaceae should be evaluated. Beta- lactamase genes and AmpC-type b-lactamase genes such as blaCTX-M and blaCMY and *aphA1* (neomycin resistance) and the specific correlation between gene mutations and growth regulatory processes should be investigated [34].

## 5. Conclusions

The present cohort study provided an overview on the rates at which antimicrobial resistance is acquired or lost under the routine management of pre-weaned dairy calves in California using *Enterobacteriaceae* as an indicator. Administering ceftiofur as a treatment to calves resulted in an increasing concentration of resistant *Enterobacteriaceae* in feces (on MacConkey infused with ceftiofur 30 µg/mL), showing an acquisition of resistance. In particular, the increasing *Enterobacteriaceae* concentration due to the treatments resulted in a peak of resistance around four days post treatment. Starting day five post treatment there was an evident decrease in the concentration of resistant *Enterobacteriaceae* reflecting loss of resistance at the 30 µg ceftiofur/mL concentration. Furthermore, our data shows that all the study calves at start of follow up shed resistant *Enterobacteriaceae* at higher levels compared to the end of follow up in untreated calves. The reasons for calves shedding resistant Enterobacteriaceae during the first days of life are not known, however this could be associated with dry cow antimicrobial therapy residue in the colostrum fed in the study calves, or simply due to colonization of the calves’ gut microbiota with resistant bacteria from the environment. The neomycin administrated to our calf cohort during the first 20 days from enrollment was associated with an increase in neomycin resistant *Enterobacteriaceae*, followed by loss of resistance immediately after the end of the neomycin administration.

## Figures and Tables

**Figure 1 microorganisms-09-02103-f001:**
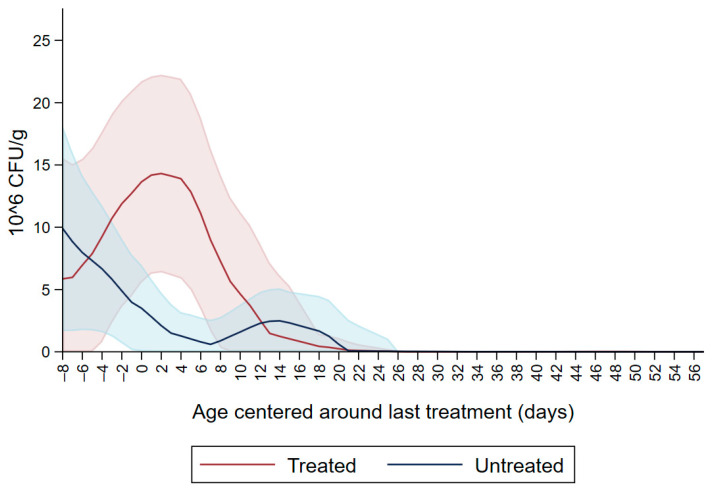
Ceftiofur resistant (≥30 µg/mL) Enterobacteriaceae concentration in fecal samples plotted by untreated and treated calves on the first 9 days with ceftiofur. The colored graphs clouds represent the respective 95% confidence intervals for treated and untreated calves.

**Figure 2 microorganisms-09-02103-f002:**
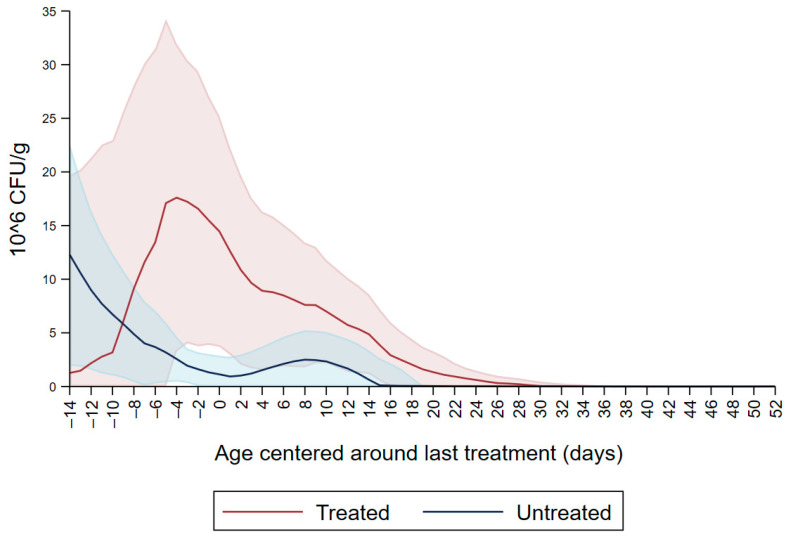
Ceftiofur resistant (≥30 µg/mL) Enterobacteriaceae concentration in fecal samples plotted by untreated and treated calves from 10 to 18 days with ceftiofur. The colored graphs clouds represent the respective 95% confidence intervals for treated and untreated calves.

**Figure 3 microorganisms-09-02103-f003:**
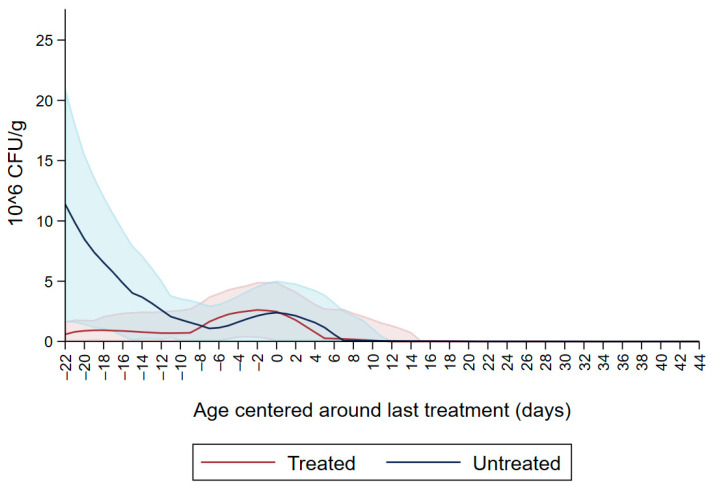
Ceftiofur -resistant (≥30 µg/mL) Enterobacteriaceae concentration in fecal samples plotted by untreated and treated calves after 19 days with ceftiofur. The colored graphs clouds represent the respective 95% confidence intervals for treated and untreated calves.

**Figure 4 microorganisms-09-02103-f004:**
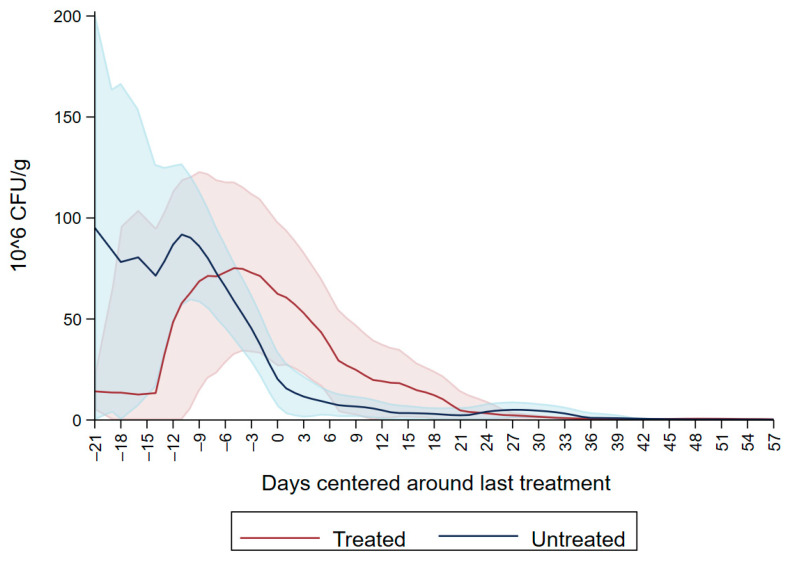
Difference in concentration of *Enterobacteriaceae* between MacConkey non-infused by antibiotics and MacConkey agar infused by ceftiofur 30 µg/mL from fecal samples collected from 33 calves followed from enrollment to 64 days age.

**Figure 5 microorganisms-09-02103-f005:**
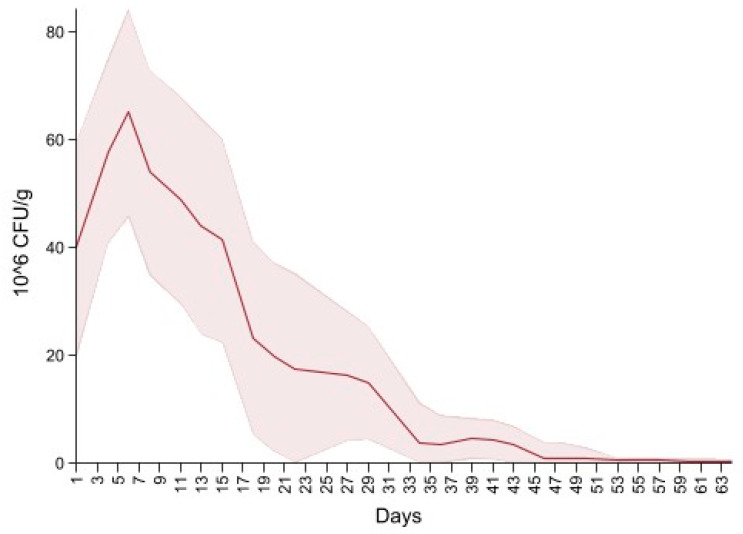
Neomycin-resistant (≥50 µg/mL) Enterobacteriaceae concentration in fecal samples collected from 33 calves from enrollment to 64. days age. All calves were fed milk replacer supplemented by neomycin for the first 20 days of life.

**Figure 6 microorganisms-09-02103-f006:**
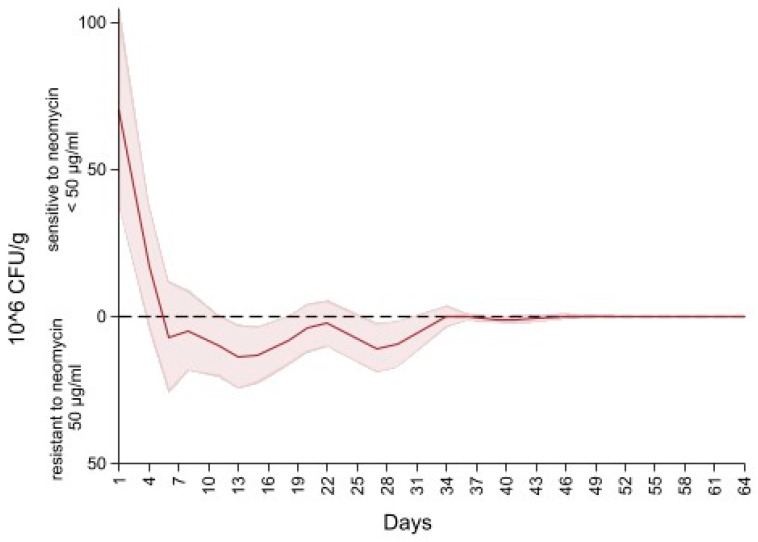
Difference in concentration of Enterobacteriaceae between MacConkey non-infused by antibiotics and MacConkey agar infused with neomycin 50 µL/mL from fecal samples collected from 33 calves followed from enrollment to 64 days age.

**Table 1 microorganisms-09-02103-t001:** Number of treated calves in a cohort of 33 calves monitored from arrival at a calf ranch after birth to 60 days of age.

Study Calf ID	Number of Ceftiofur Treatment Doses	Treatment Study Day	Disease
1	2	7, 8	Diarrhea
4	3	12, 13, 14	Diarrhea
8	9	4, 5, 6, 7, 8, 9, 12, 13, 14	Diarrhea
12	3	6, 7, 8	Diarrhea
16	3	7, 8, 9	Diarrhea
19	2	6, 7	Diarrhea
22	1	20	Diarrhea
23	4	6, 7, 8, 9	Diarrhea
24	3	5, 6, 7	Diarrhea
26	3	6, 7, 8	Diarrhea
34	1	22	Diarrhea
37	2	7, 8	Diarrhea
52	2	9, 14	Diarrhea

**Table 2 microorganisms-09-02103-t002:** Final mixed effects negative binomial model for the concentration of ceftiofur-resistant (≥30 µg/mL) Enterobacteriaceae in fecal samples collected from 33 calves followed from enrollment to 64 days of age.

		Estimate ^1^	S.E.	*p* Value	95% Confidence Interval	
					Lower Limit	Upper Limit
DPAT ^2^	Pre-treatment	referent				
	During treatment	1.52	1.276	0.231	−0.97	4.02
	Post-treatment					
	1–2	2.06	1.503	0.170	−0.88	5.01
	3–4	2.97	1.503	0.034	0.22	5.71
	5–6	0.96	1.347	0.476	−1.68	3.60
	7–8	−0.27	1.528	0.858	−3.26	2.72
	9–10	1.38	1.803	0.443	−2.15	4.91
	11–12	0.32	1.662	0.847	−2.93	3.57
	13–14	−0.01	1.589	0.992	−3.13	3.09
	15–64	1.20	1.596	0.451	−1.92	4.33
Fecal score						
	Normal—loose	referent				
	Watery	−1.68	0.90	0.06	−3.44	0.08
Intercept		18.59	0.61	0.00	17.39	19.78
Ln (dispersion coefficient)		2.63	0.05		2.52	2.75
Random effects						
	Calf (Random Intercept)	4.71	2.40		1.73	12.83
	Follow up days (Random slope)	0.10	0.02		0.06	0.18

^1^ Negative binomial model coefficient comparing the explanatory variable’s identified level to the respective referent. Coefficients are magnitudes of change on the natural logarithm scale and hence should be interpreted with the intercept and any other covariate, ^2^ Days Post Administration of Treatment (DPAT) specified Pre-treatment which are the days of follow up of untreated calves and pretreatment in treated calves; During Treatment as the days during treatment of treated calves; Post Treatment as the remaining values specify days after last treatment. Treatments refers to the parental ceftiofur hydrochloride therapy.

**Table 3 microorganisms-09-02103-t003:** Final mixed negative binomial model for the difference in the concentration of Enterobacteriaceae in MacConkey non-infused with antimicrobials and MacConkey infused with ceftiofur 30 µg/mL from fecal samples collected from 33 calves followed from enrollment to 64 days of age.

		Estimate ^1^	S.E.	*p* Value	95% Confidence Interval	
					Lower Limit	Upper Limit
DPAT ^2^	Pre-treatment	Referent				
	During treatment	0.99	0.569	0.080	−0.12	2.11
	Post-treatment					
	1–2	0.02	0.771	0.975	−1.48	1.53
	3–4	0.61	0.713	0.386	−0.77	2.01
	5–6	−0.31	0.727	0.662	−1.74	1.10
	7–8	−2.21	0.763	0.004	−3.71	−0.72
	9–10	−0.19	0.927	0.835	−2.01	1.62
	11–12	−3.30	0.836	<0.001	−4.94	−1.66
	13–14	−0.41	0.808	0.609	−1.99	1.17
	15–16	−1.17	1.044	0.260	−3.22	0.87
	17–18	−2.61	1.415	0.065	−5.38	0.16
	19–20	−1.16	0.857	0.173	−2.84	0.51
	21–22	−2.58	0.927	0.005	−4.40	−0.77
	23–24	−5.04	1.702	0.003	−8.38	−1.71
	25–26	−2.25	0.977	0.021	−4.17	−0.34
	27–28	−2.00	1.014	0.048	−3.99	−0.14
	29–30	−2.33	1.166	0.045	−4.62	−0.05
	31–32	−3.42	1.108	0.002	−5.59	−1.24
	33–34	−3.72	1.092	0.001	−5.87	−1.58
	35–36	−3.59	1.141	0.002	−5.83	−1.35
	37–38	−3.40	1.302	0.009	−5.96	−0.85
	39–40	−4.33	1.191	<0.001	−6.67	−2.00
	41–42	−3.03	1.286	0.018	−5.55	−0.51
	43–44	−5.59	1.348	<0.001	−8.23	−2.94
	45–46	−4.99	1.372	<0.001	−7.68	−2.30
	47–48	−4.04	1.412	0.004	−6.81	−1.27
	49–50	−4.93	1.467	0.001	−7.80	−2.05
	51–52	−3.42	1.773	0.053	−6.90	0.04
	53–54	−4.14	1.637	0.011	−7.34	−0.93
	55–56	−5.26	1.617	0.001	−8.43	−2.09
	57–58	−4.79	2.221	0.031	−9.14	−0.44
Intercept		17.82	0.179	<0.001	17.47	18.17
Ln (dispersion coefficient)		1.48	0.04		1.40	1.56
Random effects						
	Calf (Random intercept)	0.114	0.174		0.005	2.26
	Follow up days (Random slope)	0.0057	0.0014		0.0034	0.0095

^1^ Negative binomial model coefficient comparing the explanatory variable’s identified level to the respective referent. Coefficients are magnitudes of change on the natural logarithm scale and hence should be interpreted with the intercept and any other covariate, ^2^ Days Post Administration of Treatment (DPAT) specified Pre-treatment which are the days of follow up of untreated calves and pretreatment in treated calves; During Treatment as the days during treatment of treated calves; Post Treatment as the remaining values specify days after last treatment. Treatments refers to the parental ceftiofur hydrochloride therapy.

**Table 4 microorganisms-09-02103-t004:** Final mixed negative binomial model for the concentration of neomycin-resistant (≥50 µg/mL) Enterobacteriaceae in fecal samples collected from 33 calves followed from enrollment to 64 days of age.

		Estimate ^1^	S.E.	*p* Value	95% Confidence Interval	
					Lower Limit	Upper Limit
Sampling days ^2^	During treatment					
	1	Referent				
	4	2.44	0.578	0.000	1.31	3.58
	6	1.46	0.563	0.009	0.36	2.57
	8	2.01	0.579	0.001	0.87	3.14
	11	0.88	0.567	0.121	−0.23	1.99
	13	1.70	0.580	0.003	0.56	2.84
	15	0.43	0.574	0.446	−0.68	1.56
	18	1.16	0.586	0.047	0.15	2.31
	20	−1.88	0.567	0.001	−3.00	−0.77
	Post-treatment					
	22	0.82	0.596	0.169	−0.34	1.98
	27	0.55	0.617	0.365	−0.65	1.76
	29	−0.62	0.601	0.302	−1.80	0.55
	34	−1.03	0.595	0.083	−2.19	0.13
	36	−0.89	0.663	0.176	−2.19	0.40
	39	−1.69	0.592	0.004	−2.85	−0.53
	41	−0.80	0.685	0.240	−2.14	0.53
	43	−2.28	0.610	0.000	−3.48	−1.09
	46	−2.78	0.610	0.000	−3.98	−1.59
	48	−2.55	0.613	0.000	−3.75	−1.34
	50	−2.14	0.625	0.001	−3.37	−0.92
	53	−5.00	0.625	0.000	−6.22	−3.77
	55	−2.96	0.665	0.000	−4.26	−1.65
	57	−4.42	0.626	0.000	−5.65	−3.19
	60	−4.22	0.662	0.000	−5.52	−2.92
	62	−3.90	0.651	0.000	−5.17	−2.62
	64	−4.38	0.658	0.000	−5.67	−3.09
Intercept		16.03	0.452	0.000	15.15	16.92
Ln (dispersion coefficient)		1.38	0.04		1.30	1.46
Random effects						
	Calves’ age in days (Random intercept)	0.000	0.000		0.000	0.001
	Id (Random slope)	0.405	0.235		0.129	1.265

^1^ Negative binomial model coefficient comparing the explanatory variable’s identified level to the respective referent. Coefficients are magnitudes of change on the natural logarithm scale and hence should be interpreted with the intercept and any other covariate, ^2^ Sampling days are the days of follow up of the calves which all received neomycin treatment into the milk replacer.

**Table 5 microorganisms-09-02103-t005:** Prediction estimates based on mixed effect negative binomial models of ceftiofur resistant counts of Enterobacteriaceae (≥30 µg/mL). Calculated based on their respective models, bracketed values are the 95% confidence intervals.

	Enterobacteriaceae (×,10^6^) per Gram of Feces at ≥30 µg/mL Ceftiofur			
Time	Normal/Loose Feces		Watery Feces	
Pre-treatment/Untreated	119	(0–260)	22.1	(0–66)
During Treatment	547	(0–1970)	102	(0–372)
Days Post-Treatment:				
1–2	936	(0–3750)	174	(0–811)
3–4	2310	(0–8810)	431	(0–1890)
5–6	310	(0–1170)	57.7	(0–252)
7–8	90.3	(0–369)	16.8	(0–77)
9–10	473	(0–2190)	88.1	(0–447)
11–12	164	(0–719)	30.5	(0–139)
13–14	117	(0–490)	21.7	(0–104)
15–64	690	(0–1750)	73.6	(0–365)

**Table 6 microorganisms-09-02103-t006:** Prediction estimates based on mixed effect negative binomial models of ceftiofur sensitive counts of Enterobacteriaceae (<30 µg/mL). Calculated based on their respective models, bracketed values are the 95% confidence intervals.

Time	Enterobacteriaceae (×,10^6^) per Gram of Feces at <30 µg/mL Ceftiofur	
Pre-treatment/Untreated	55.2	(35.8–74.5)
During Treatment	149	(0–312)
Days Post-Treatment:		
1–2	56.5	(0–141)
3–4	102	(0–245)
5–6	40.2	(0–96.9)
7–8	6	(0–15.0)
9–10	45.5	(0–128)
11–12	2.02	(0–5.3)
13–14	36.2	(0–94.2)
15–16	17	(0–51.9)
17–18	4	(0–15.3)
19–20	17.2	(0–46.2)
21–22	4.1	(0–11.7)
23–24	0.35	(0–1.53)
25–26	5.7	(0–16.9)
27–28	7.4	(22.4)
29–30	5.3	(0–17.6)
31–32	1.8	(0–5.7)
33–34	1.3	(0–4.1)
35–36	1.5	(0–4.9)
37–38	1.8	(0–6.5)
39–40	0.72	(0–2.4)
41–42	2.6	(0–9.4)
43–44	0.20	(0–0.75)
45–46	0.37	(0–1.3)
47–48	0.96	(0–3.6)
49–50	0.39	(0–1.5)
51–52	1.7	(0–8.0)
53–54	0.87	(0–3.7)
55–56	0.28	(0–1.2)
57–58	0.45	(0–2.4)

**Table 7 microorganisms-09-02103-t007:** Prediction estimates based on mixed effect negative binomial models of neomycin resistant counts of Enterobacteriaceae (≥50 µg/mL). Calculated based on their respective models, bracketed values are the 95% confidence intervals.

Time	Enterobacteriaceae (×,10^6^) per Gram of Feces at ≥50 µg/mL Neomycin	
During Treatment		
1	9.2	(10.4–17.4)
4	107	(27.1–186)
6	40	(10.6–69.4)
8	69	(16.7–121)
11	22.3	(5.1–39.4)
13	50.6	(10.1–91.2)
15	14.3	(3–25.6)
18	29.6	(5.8–53.3)
20	1.3	(0.34–2.4)
Days Post-Treatment:		
22	21	(4.2–37.7)
27	16.1	(1.9–30)
29	4.9	(0.71–9.2)
34	3.2	(0.60–5.9)
36	3.7	(0.0079–7.5)
39	1.6	(0.30–3)
41	4.1	(0–8.5)
43	0.93	(0.12–1.7)
46	5.6	(0.0730–10.6)
48	0.71	(0.91–1.3)
50	1	(0.10–2.05)
53	0.06	(0.0051–0.1186)
55	0.47	(0–0.96)
57	0.11	(0.0091–0.2125)
60	0.13	(0–0.27)
62	0.18	(0.0021–0.3713)
64	0.11	(0–0.22)

## Data Availability

The herd owners did not consent to sharing the raw data; as such, raw data from this study is not available.

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
