# Peer review of "Estimating the Rates of Acquisition and loss of Resistance of *Enterobacteriaceae* to Antimicrobial Drugs in Pre-Weaned Dairy Calves"

_microorganisms, 2021, doi:10.3390/microorganisms9102103_

Round 1
Reviewer 1 Report
The manuscript entitled “
Estimating the rate of acquisition and loss of resistance of Enterobacteriaceae to antimicrobial drugs in pre-weaned dairy calves.” is discussing an interesting issue: The antibiotic resistance in preweaned calves to two important antimicrobial drugs. Although it was not discussed in this study, this problem can be of a high zoonotic importance. The calves can act as a zoonotic reservoir for human infections and transmission of antibiotic resistance associated genes causing treatment failure in humans.
The manuscript is concise, and provides an overview on the current status in such dramatically increasing emergence. The experimental results are adequately described, their interpretation is reasonable and organized discussion. However, I have some few comments, to mention:
- Some sentences throughout the whole manuscript are either too long or too complicated to be understood. They have to be rephrased please to make. I did not mention some because they are throughout the whole manuscript.
- Is the study including 33 or 55 calves? If only 33 were selected, is this due to diarrhea symptoms? Please describe this point in details.
- Is ceftiofur was prescribed by the veterinarian in farms for treatment of the diarrhea cases among calves or that was the decision of the authors?
- Please also identify in details the idea behind adding neomycin sulfate to milk replacer, is it only for prevention of infections? Is it a common regimen in dairy farms in USA?
- For how long were the calves fed with colostrum? Please identify. Also, is weaning age normally in farms at 2 months or the calves were only followed for this time period?
- From oy point of view, it is a great point of weakness, when the authors do not include any genotypic identification for resistance associated genes at all. It was stated in discussion that genotypic identification will be tested separately. However, only phenotypic identification is not enough for judgment in all cases. Most of time, discrepancies are recorded between phenotypic and genotypic resistance identification. This phenomena is attributed to many factors, such as absence of expression of some resistance genes or Efflux pumps.
- It would be great if the authors could include at least minimum limit of genotypic identification of resistance associated genes.
I believe that although the soundness and novelty of the study, some important perspectives are missed in the study.
I would recommend major revisions before publishing the manuscript.
Reviewer 2 Report
The authors have shown that antibiotic treatment selects resistance, however the taxonomic classification of family alone, in my opinion, belittles the significance of this manuscript.
In the introduction the authors describe the role of Escherichia coli, however it is strange that then they did not think of a speciation rather than stop at the family level.
An unclear point is the MIC of E. coli strain 25922 for ceftiofur. This is quality control and normally sensitive strain. Did the authors induce resistance? Did the authors test whether the same susceptibility levels were demonstrable in Mueller Hinton? Did the authors verify the MIC for ceftiofur of strain 25922? In fact, the meaning of this experiment is not clear.
Although I can understand that they wanted to make MacConkey even more selective, in the sense of selecting only resistant strains, the selection of 30 μg/ml is not acceptable, since the break point for the molecule is ≥ 8 μg/ml to define strains as resistant.
Did the authors verify, with standardized techniques, that the isolates were really resistant? Did they verify with molecular techniques which were the determinants of resistance of the isolates?
Reviewer 3 Report
This study evaluated the persistence of "resistant" Enterobacteriaceae in preweaned dairy calves after exposure to antimicrobials. The subject matter is timely and in general, the manuscript is well written. There are several study design limitations that should be addressed and re-reviewed prior to publication. I think the assessment of a neomycin treatment (as all calves were fed for a defined period of time) makes great sense. The data surrounding ceftiofur treatment is less defendable because the frequency and timing of the antibiotic treatment is not clear. Also, were the 33 calves included given antimicrobials other than ceftiofur and neomycin? If that is the case, co-selection of "resistance" would be important. Please provide more complete treatment records to assess.
In the introduction you make a strong case for dairy calves exposure to neomycin, however, I didn't feel like there was a strong, if any case made for ceftiofur - other than the convenience of treatment methods employed in this study it isn't well justified why that treatment was chosen. Please expand.
Although statistically you appear to account for different sources of calves, that seems like something that should have been randomized and blocked at the beginning of the study. Was that done and if so, can you please include in the methods.
It isn't clear why the 33 calves were used when 55 calves were enrolled in the study. Were the other 22 not given ceftiofur so excluded? Which number was your power based on? Enrolling animals you don't evaluate might be okay, but it isn't well explained.
I don't believe that the use of antibiotic "susceptibility" is appropriate for your screening tool. You were probably, at best, assessing antimicrobial tolerance to the level of antimicrobial selected, but susceptibility is a term used to reflect a clinical response based on established breakpoints and those were not used here. I would strongly suggest changing the terms susceptible and resistance throughout when that really isn't what you measured.
Line 291: why weren't the clinical breakpoints for ceftiofur or neomycin used in your supplement? The method by which you chose to determine the concentration of antibiotic evaluated very few isolates and could mean absolutely nothing for achievable concentrations at the site (GI tract in this case).
I don't know that the data in the table 1 clearly describes which animals were treated, and if those animals were then sampled. Additionally, I am having a hard time wrapping my head around animals being treated at different times in the study period and then arbitrarily picking some of those animals to sample at variable days after that treatment? I think some more description or justification is warranted so this is better understood.
Freezing samples prior to quantification is really not best practice. Were all samples held for the same amount of time in the freezer prior to analysis? Were earlier samples held longer and what might the impact of this be?
Minor comments:
Line 145: Commensal bacteria from cattle are less resistant than other species - this is a broad generalization and the reference (only 1 provided) is from a geographically limited area. I would restructure the sentence or remove it completely as it does not support the justification for your work.
Line 148: pre-unweaned - are calves not unweaned or pre-weaned? I'm not sure you mean both. Additionally, your formatting for naming convention changes through the document.
You make the case for the Aust study (reference 22) showing transient shedding of resistant organisms after treatment. Please consider distinguishing or building upon this to justify your work (which seems similar with somewhat identical findings).
Round 2
Reviewer 1 Report
Hereby, I accept the current form of the manuscript to be published.
Best of luck
Reviewer 2 Report
In my opinion, the authors' answers are not convincing. No answers have been given. The manuscript in this form is unacceptable
Author Response
Please see attached response file.
